# Modular access to chiral bridged piperidine-γ-butyrolactones via catalytic asymmetric allylation/*aza*-Prins cyclization/lactonization sequences

Cong Fu[1,2,4], Ling He[1,4], Hui Xu[3,4], Zongpeng Zhang[1], Xin Chang[1], Yanfeng Dang[3] ✉, Xiu-Qin Dong [1] ✉ & Chun-Jiang Wang[1,2] ✉

Chiral functionalized piperidine and lactone heterocycles are widely spread in natural products and drug candidates with promising pharmacological properties. However, there remains no general asymmetric methodologies that enable rapid assemble both critical biologically important units into one three-dimensional chiral molecule. Herein, we describe a straightforward relay strategy for the construction of enantioenriched bridged piperidine-γ-butyrolactone skeletons incorporating three skipped stereocenters via asymmetric allylic alkylation and *aza*-Prins cyclization/lactonization sequences. The excellent enantioselectivity control in asymmetric allylation with the simplest allylic precursor is enabled by the synergistic Cu/Ir-catalyzed protocol; the success of *aza*-Prins cyclization/lactonization can be attributed to the pivotal role of the ester substituent, which acts as a preferential intramolecular nucleophile to terminate the *aza*-Prins intermediacy of piperid-4-yl cation species. The resulting chiral piperidine-γ-butyrolactone bridged-heterocyclic products show impressive preliminary biological activities against a panel of cancer cell lines.

Searching for efficient synthetic protocols to construct enantioenriched heterocyclic molecular architectures with three-dimensionally complex geometries is a fundamental task in synthetic and medicinal chemistry since saturation and complexity frequently provide enhanced biological activities and improved clinical success[1–3]. In this context, ubiquitous piperidines and γ-butyrolactones, especially in optically active forms, represent the crucial scaffolds in numerous molecules of pharmaceutical interest that exhibit an impressive range of biological profiles[4] (Fig. 1a). Piperidine, as a privileged motif, is the most commonly used nitrogen heterocycle among the small-molecule drugs approved by FDA[5]. For example, piperidine-containing cocaine has the latent abilities to suppress the reuptake of DA (dopamine) and 5-HT (serotonin), which

is an in vivo ligand for muscarinic acetylcholine receptors[6,7]. The anti-allergic drug, levocabastine[8], contains two adjacent stereocenters on the piperidine ring. Meanwhile, piperidines are also versatile precursors for the privileged chiral ligands and organocatalysts in asymmetric synthesis[9,10]. On the other hand, γ-butyrolactone rings broadly exist in many steroids, sesquiterpene lactones, macrocyclic antibiotics, alkaloids, and pharmaceuticals[11]. The simplest γ-butyrolactone itself is an approved prodrug for narcolepsy[12], and antiviral, anti-inflammatory, cytotoxic, antimicrobial, and phytotoxic properties are generally ascribed to γ-lactone moieties.

The biological importance and synthetic utility of piperidine and γ-butyrolactone moieties, especially in their enantioenriched form, has

[1]College of Chemistry and Molecular Sciences, Wuhan University, Wuhan 430072, China. [2]State Key Laboratory of Elemento-organic Chemistry, Nankai University, Tianjin 300071, China. [3]Tianjin Key Laboratory of Molecular Optoelectronic Sciences, Department of Chemistry, Tianjin University, Tianjin 300072, China. [4]These authors contributed equally: Cong Fu, Ling He, Hui Xu. ✉e-mail: yanfeng.dang@tju.edu.cn; xiuqindong@whu.edu.cn; cjwang@whu.edu.cn

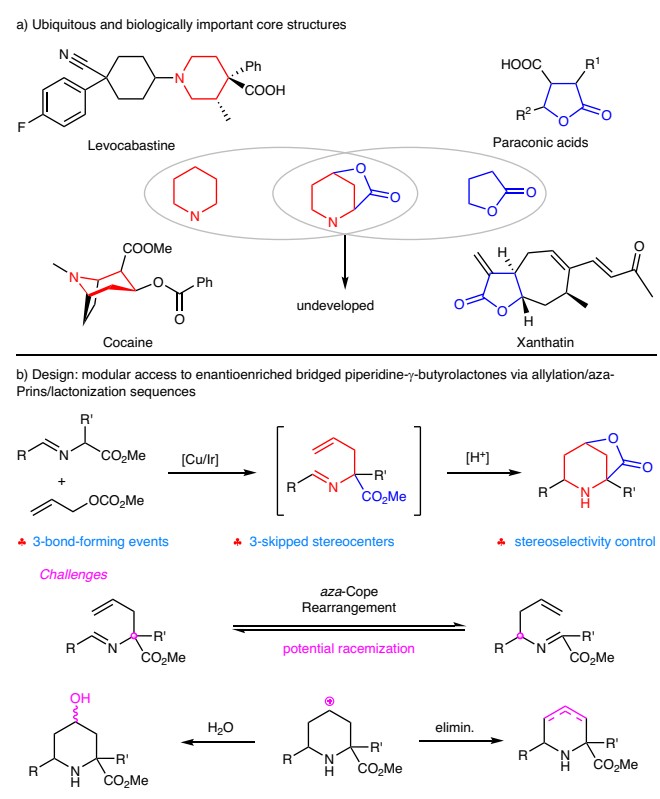

**Fig. 1 | Background and our design. a** Ubiquitous piperidine and γ-butyrolactone structural core; **b** Design plan for the catalytic asymmetric synthesis of bridged piperidine-γ-butyrolactone skeletons.

prompted the extensive development of methodologies for their asymmetric synthesis in the past decades[13–19]. Except for the asymmetric [4 + 2]-cycloaddition between the dienophiles and *N*-protected imine[20,21], *aza*-Prins cyclization has been recognized as a streamlined and reliable protocol to build piperidine from readily available materials[22,23]. The *aza*-Prins process typically involves the reaction of homoallylic amines with aldehydes (or corresponding aldimines directly) in the presence of either a protonic or Lewis acid to deliver the intermediacy of piperidyl cations, which are subsequently trapped by nucleophiles, such as water, (pseudo)halogen, nucleophilic solvents, Friedel-Crafts type, and Ritter type terminations. However, many of them are often accompanied by complicated processes and/or poor diastereoselectivity, and the construction of enantioenriched piperidines via *aza*-Prins cyclization in a stereoselective manner is scarcely accomplished except in sporadic examples with chiral auxiliary groups[24]. There remains no general asymmetric variant of the *aza*-Prins reaction for constructing enantioenriched piperidines. In comparison, remarkable progress in the construction of enantioenriched γ-butyrolactones has been achieved in recent decades, including functionalization of furanone derivatives, halolactonization, NHC-promoted [3 + 2] annulation, and metal-carbene transformation[17]. We envisioned that an appropriate combination of the two critical biologically important units into one bridged piperidine-γ-butyrolactone motif, a three-dimensional molecule with saturation and complexity, would introduce some benefits and is reckoned to find potential applications in drug discovery and medicinal chemistry. In stark contrast to the well-established methodologies to approach either piperidine or γ-butyrolactone-containing compounds, to our knowledge, an efficient protocol that enables the simultaneous construction of the bridged piperidine-γ-butyrolactone structural skeletons in stereoselective manner remains elusive and is utmost appealing.

In continuation of our research on azomethine ylides[25] and asymmetric bimetallic catalysis[26–62], we envisaged that a dual metal-catalyzed asymmetric allylic alkylation of racemic α-substituted aldimine esters would provide a direct and efficient approach to access the key intermediacy of enantioenriched aldimine esters possessing a delicate pendant allylic unit, which are eligible precursors for the following cascade asymmetric *aza*-Prins annulation/lactonization (Fig. 1b). We supposed that the allylated aldimine ester would be activated by Brønsted or Lewis acid to furnish a more electron-deficient iminium ion, which could undergo π-nucleophilic attack by the tethered allylic group. Terminating the newly forming cyclic carbocation by the ensuing intramolecular nucleophile attack of the ester group would afford a bridged piperidine-γ-butyrolactone with sequential formation of two additional stereogenic centers, which were presumed to be precisely controlled by asymmetric intramolecular induction via a highly ordered six-membered chair-like transition state. In this manner, a sequential reaction with readily-available starting materials, merging catalytic asymmetric allylic alkylation, *aza*-Prins cyclization, and lactonization, would provide a concise protocol to construct the bridged piperidine-γ-butyrolactones adorned with three stereogenic centers that are otherwise inaccessible (Fig. 1b). However, there are several challenges in this reaction design: 1) Despite significant advances in transition-metal catalyzed asymmetric allylic alkylation[63–67], less satisfactory enantioselectivities (generally <91% ee) were generally reported when α-substituted aldimine ester reacted with the simplest allylic precursors[36,49,68] in comparison with 1-substituted counterparts; 2) To find appropriate reaction condition that would promote *aza*-Prins cyclization while suppressing the undesired *aza*-Cope rearrangement[69], the latter would decay the stereo-integrity of the following stereogenic center and thus result in poor diastereo-/enantioselectivity control of the additional stereogenic centers in the subsequent *aza*-Prins/lactonization cascade; 3) The competitive other nucleophilic terminators, such as solvent or incidental water *vs* intramolecular nucleophilic lactonization, or the disturbance caused by the elimination of the resultant piperid-4-yl cation[22,23].

Herein, we documented the development, the diastereo-/enantioselectivity control, the substrate generality, and synthetic elaborations of the synergistic Cu/Ir-catalyzed asymmetric allylation/*aza*-Prins cyclization/lactonization of aldimine esters with allylic carbonate. The current protocol provides a general process for efficiently preparing a variety of enantioenriched heterocyclic bridged piperidine-γ-butyrolactones. The exclusive stereochemical outcome of the whole sequential process is remarkable given three sequential C–C/C–C/C–O bond-forming events accompanied by the sequential formation of three skipped stereogenic centers. Preliminary biological activity studies demonstrated that the enantioenriched bridged heterocycles are potential anticancer agents.

## Results

### Condition optimization

Initially, our investigations were focused on asymmetric allylic alkylation reaction between allyl carbonate **1** and alanine-derived aldimine ester **2a**. In consideration of that metallated azomethine ylides are kind of stereochemically tunable nucelophiles[70,71] and no allyl stereogenic center will be generated in the expected coupling product, we firstly studied the feasibility of the reaction with the dual chiral Cu/achiral Pd co-catalyst system established in our previous work[49]. The intermediacy of α-allylic substituted aldimine ester **3a** was obtained in 56% yield and unsatisfactory 80% ee in the presence of [Cu(I)/(S,$S_p$)-**L1** + Pd(PPh$_3$)$_4$] catalyst combination, together with Cs$_2$CO$_3$ as the base and CH$_2$Cl$_2$ as the solvent (Table 1, entry 1). Subsequently, we examined the reactivity and stereoselectivity control with different chiral Phosferrox ligands (**L2**–**L4**) for copper(I) complex, and no further improvement was observed (entries 2–4).

## Table 1 | Optimization of reaction conditions[a]

| entry | Cu-complex | Pd- or Ir-complex | PG | Yield (%)[b] | ee (%)[c] |
|---|---|---|---|---|---|
| 1 | [Cu]/$(S,S_p)$-L1 | Pd(PPh$_3$)$_4$ | Boc | 56 | 80 |
| 2 | [Cu]/$(S,S_p)$-L2 | Pd(PPh$_3$)$_4$ | Boc | 47 | 77 |
| 3 | [Cu]/$(S,S_p)$-L3 | Pd(PPh$_3$)$_4$ | Boc | 58 | 69 |
| 4 | [Cu]/$(S,S_p)$-L4 | Pd(PPh$_3$)$_4$ | Boc | 42 | 74 |
| 5[d] | [Cu]/$(S,S_p)$-L1 | [Pd]/$(S,S_p)$-L1 | Boc | 45 | 4 |
| 6 | [Cu]/$(S,S_p)$-L1 | [Ir]/$(S,S,S)$-L5 | Boc | 77 | 99 |
| 7 | [Cu]/$(S,S_p)$-L2 | [Ir]/$(S,S,S)$-L5 | Boc | 74 | 98 |
| 8 | [Cu]/$(S,S_p)$-L3 | [Ir]/$(S,S,S)$-L5 | Boc | 74 | 99 |
| 9 | [Cu]/$(S,S_p)$-L4 | [Ir]/$(S,S,S)$-L5 | Boc | 75 | 99 |
| 10 | [Cu]/$(S,S_p)$-L1 | [Ir]/$(S,S,S)$-L5 | CO$_2$Me | 94 | >99 |
| 11 | [Cu]/$(S,S_p)$-L1 | [Ir]/$(S,S,S)$-L5 | Ac | 84 | 93 |
| 12 | [Cu]/$(R,R_p)$-L1 | [Ir]/$(S,S,S)$-L5 | CO$_2$Me | 87 | −94 |
| 13 | [Cu]/$(S,S_p)$-L1 | [Ir]/rac-L5 | CO$_2$Me | 82 | 95 |
| 14[e] | [Cu]/$(S,S_p)$-L1 | [Ir]/$(S)$-L6 | CO$_2$Me | 93 | 75 |
| 15[e] | [Cu]/$(R,R_p)$-L1 | [Ir]/$(S)$-L6 | CO$_2$Me | 90 | −64 |
| 16 | [Cu]/ DPE-Phos | [Ir]/$(S,S,S)$-L5 | CO$_2$Me | 73 | 53 |
| 17 | - | [Ir]/$(S,S,S)$-L5 | CO$_2$Me | trace | - |
| 18 | [Cu]/$(R,R_p)$-L1 | - | CO$_2$Me | trace | - |

[a]All reactions were carried out with 0.3 mmol 1, 0.2 mmol 2a, 0.3 mmol Cs$_2$CO$_3$ in 2 mL of CH$_2$Cl$_2$ for 10-14 h. Cu(I) = Cu(MeCN)$_4$BF$_4$. Ir(I) = [Ir(cod)Cl]$_2$.
[b]Isolated yields.
[c]Ee was determined by chiral HPLC analysis.
[d]Cu(OTf)$_2$ and [Pd(allyl)Cl]$_2$ were used as the metal precursors with THF as the solvent.
[e]The ratio of Ir/$(S)$-L6 is 1:2.

Attempts with the identical chiral ligands for both copper and palladium complex also proved to be futile (entry 5). It was occurred to us that the enantioselectivity might be significantly improved through replacing the achiral palladium complex in the dual Cu/Pd catalyst combination with the chiral Ir/chiral phosphoramidite complex. However, since Ir-catalyzed asymmetric allylic substitution reactions mainly provide branched allylated products incorporating an allyl stereogenic center, it is not curious that there are no previous examples of Ir-catalyzed asymmetric allylic substitution reaction using the simplest allylic precursor[66]. To our delight, switching Pd catalysts to Ir(I)/$(S,S,S_a)$-L5[72] complex enabled 77% isolated yield with excellent stereoselectivity control (99% ee, entry 6). Further modification of different substituent groups on the oxazoline ring of Phosferrox ligands revealed an insignificant influence on the reaction outcomes (entries 7–9). Replacement of leaving group OBoc with OCO$_2$Me or OAc in allylic carbonates showed a considerable improvement of the yield, and allylic methyl carbonate 1b gave the

best result in 94% yield with >99% ee (entries 10 and 11). With the diastereomeric set of [Cu(I)/$(R,R_p)$-L1 + Ir(I)/$(S,S,S_a)$-L5] catalyst combination, the reaction proceeded smoothly to afford the desired compound ent-3a in 87% yield and an inferior level of enantioselectivity albeit with the opposite sense of asymmetric induction (entry 12). In addition, unsatisfactory asymmetric induction was observed when the combined [Cu(I)/$(S,S_p)$-L1 + Ir(I)/rac-L5] was employed in this allylation (entry 13). Those experimental results indicate that the set of [Cu(I)/$(S,S_p)$-L1 + Ir(I)/$(S,S,S_a)$-L5] are the matched catalyst combination, and the configuration of the allylation products was determined mainly by chiral copper(I) complex. To the best of our knowledge, the current synergistic Cu/Ir-catalysis protocol not only solves the intractable question of unsatisfactory stereochemical control with Cu/Pd-catalysis, but also represents a rare example of asymmetric allylation with the simplest electrophilic π-allyl-Ir(III) species. In addition, Ir(I)/$(S)$-L6 was also applied into this catalytic system, and product 3a was obtained in good yield albeit with

unsatisfactory enantioselectivity (entries 14 and 15). The indispensable and significant role of the chiral copper complex was further confirmed by the control experiment with the set of [Cu(I)/DPE-Phos + Ir(I)/($S,S,S_a$)-**L5**] complex, which gave rise to unsatisfactory results on the reactivity and enantioselectivity control (73% yield, 53% ee, entry 16). Further control experiments revealed that only trace amount of product could be observed in the absence of either of the metal catalysts (entries 17 and 18).

After exquisite construction of the *N*-quaternary allylic stereogenic center, we proceed to investigate reaction parameters of the following *aza*-Prins cyclization/lactonization cascade, including Lewis acid/Brønsted acids, solvent and reaction temperature (see Supplementary Information for details). The optimized conditions were concluded as follow: 3.5 equiv TfOH in MeCN (0.1 M) at 40 °C for three days, under which the bridged lactone **4a** was obtained in 96% yield with 94% ee as a single observable diastereomer. To be noted, an appropriate reaction temperature that enables *aza*-Prins cyclization/lactonization smoothly but would not erode the stereo-integrity of the initially-formed *N*-quaternary allylic stereogenic center is crucial for this cascade process since the ee value of the product decreased significantly along with the elevated temperature albeit with the maintained excellent diastereoselectivity. This slight erosion of enantioselectivity (reduced from 99% ee to 94% ee) might be attributed to the reversible *aza*-Cope rearrangement, in which the preconstructed *N*-quaternary carbon center partially underwent bond rupture and re-bonding process and thus resulting in slightly racemization of the initially-formed quaternary stereogenic center. The absolute configuration of the adduct **4a** was unambiguously determined as 1*S*,3*S*,5*R* by X-ray diffraction analysis (Fig. 2). The exclusive diastereoselectivity observed in this *aza*-Prins cyclization/lactonization under the optimized reaction conditions may be attributed to a six-membered chair-like transition state (Zimmerman-Traxler model), in which both aryl and methyl groups in allylation intermediate are placed at the equatorial positions with respect to the newly forming piperidine ring, as well as the axial approach of the tethered nucleophile (ester) to the resultant piperid-4-yl cation to forge the C-O bond with the releasing of methyl cation.

## Substrate scope

We then commence inspecting the compatibility of aldimine ester. When the ester group of aldimine ester was switched to ethyl ester, product **4a** was obtained in moderate yield with unsatisfactory enantioselectivity (Table 2, entry 2). Bulky *tert*-butyl aldimine ester was further examined, and the desired product could be accessible in almost similar results (entry 3), and methyl aldimine ester was selected for subsquent investigation due to the high cost of *tert*-butyl aldimine esters. As shown in Table 2, electron-neutral (**2h**), electron-deficient (**2a**–**2g**), and electron-rich substituents (**2i**-**2m**) on the phenyl ring all tolerated well to deliver **4a**-**4m** with moderate to high yields

and excellent stereoselectivity control (41%-96% overall yields, >20:1 dr, 88%-97% ee, Table 2, entries 3–15). Importantly, this methodology exerts excellent functional group tolerance, such as halogen, trifluoromethyl, cyano, ester, nitro, sulfone, alkyl, and alkoxyl groups all left unscathed. Moreover, the positions (*ortho*-, *meta*- or *para*-) of the substituted group on the arene ring had negligible influence on the reactivity and selectivity. In addition, the *ortho*-substituted substrates **2g**, **2k**, and **2q** with steric hindrance were also favorable reaction partners to deliver the bridged lactones in 65–70% yields, 88–96% ee, and >20:1 dr (Table 2, entries 9, 13 and 19). The aldimine esters embedded with fused 1-naphthyl (**2n**), 2-naphthyl (**2o**), heteroaryl thienyl group (**2p**), pyridyl (**2q**), or indolyl group (**2r**) also proceeded smoothly, and the respective products were delivered with comparable levels of reaction results (Table 2, entries 16-20). It is noteworthy that some electron-enriched substrates with low reactivity (**2i**, **2l**, **2m**, **2p**, and **2r**) could be rectified by elevating the reaction temperature to 50 °C, and good to high conversions to the corresponding lactones were achieved with comparable ee values. In addition, the challenging alkyl aldimine **2 s** was also a suitable substrate for this transformation, affording the cyclic product **4s** with moderate yield and good enantioselectivity (Table 2, entry 21). Subsequently, the α-substituent type of aldimine ester was also evaluated. Various substituents, including linear or branched alkyl (**2t**, **2u**), benzyl (**2v**), phenyl (**2w**), aliphatic ester (**2x**), and thioether (**2y**), were all compatible, giving a wide range of α-disubstituted bridged piperidine-γ-butyrolactone derivatives in descent yields with excellent stereoselectivity (Table 2, entries 22–27). Due to the steric hindrance, no reaction occurred with the aldimine esters derived from valine (R′ = *i*Pr) and unnatural *tert*-leucine (R′ = *t*Bu) under the standard reaction conditions.

## Scale-up experiment and synthetic application

To further showcase the synthetic utility of this methodology, a gram-scale synthesis of **4a** was performed (Fig. 3, upside). It is worth mentioning that the transformation could be realized through a simple filtration and solvent exchange after the initial allylation step, and the desired adduct could be obtained in high yield with maintained diastereoselectivity and enantioselectivity (89% overall yield, >20:1 dr, 94% ee). Reduction of **4a** with LiAlH₄ afforded the amino diol **5** bearing a 4-hydroxypiperidine core, an essential scaffold in extensive drug candidates and natural products possessing extraordinary biological activities[73]. Subjecting **4a** with DIBAL-H followed by Horner-Wadsworth-Emmons olefination and intramolecular *aza*-Michael addition, *aza*-bicyclo heptane **6** bearing four stereogenic centers was obtained in good overall yield with excellent diastereoselectivity control (Fig. 3, downside).

## Computational mechanistic studies

In consideration that the simplest allyl carbonate has never been used before as an electrophilic π-allyl-Ir(III) precursor in Ir-catalyzed asymmetric allylic substitution reactions[66], DFT computational studies have been conducted to gain insights into the mechanism of the successive asymmetric allylation/*aza*-Prins cyclization/lactonization and shed some light on the origins of stereoselectivity in Cu/Ir-catalyzed asymmetric allylation with the simplest allyl carbonate and the following two consecutive annulations. The full sequential reaction pathway was explored with the model reaction between allylic carbonate **1b** and aldimine ester **2a** with the set of [Cu(I)/($S,S_P$)-**L1** + Ir(I)/($S,S,S_a$)-**L5**] catalysts. Mechanistically, the coordination of aldimine ester (**2a**) to Cu(I)/**L1** forms the chiral nucleophilic Cu(I)/**L1**-azomethine ylide species via deprotonation (see Fig. 4a and Supplementary Fig. 5). The Cu(I)/**L1**-ylide has two isomers owing to the asymmetric nature of chiral **L1** ligand. The ylide **Cu-AY** is more stable than its isomer **Cu-AY′** by 3.3 kcal/mol mainly because the

**Fig. 2 | Condition optimization and explanation.** Optimal reaction conditions for *aza*-Prins cyclization/lactonization and rationale for enantioselectivity deterioration.

**Table 2 | Substrate scope of aldimine 2[a]**

| entry | R | R' | R'' | prod. | yield (%)[b] | ee (%)[c] |
|---|---|---|---|---|---|---|
| 1 | 4-ClC$_6$H$_4$ | Me | Me | **4a** | 90 | 94 |
| 2 | 4-ClC$_6$H$_4$ | Me | Et | **4a** | 74 | 90 |
| 3 | 4-ClC$_6$H$_4$ | Me | tBu | **4a** | 83 | 94 |
| 4 | 4-CF$_3$C$_6$H$_4$ | Me | Me | **4b** | 73 | 95 |
| 5 | 4-CNC$_6$H$_4$ | Me | Me | **4c** | 93 | 95 |
| 6 | 4-MeO$_2$CC$_6$H$_4$ | Me | Me | **4d** | 96 | 96 |
| 7 | 3-NO$_2$C$_6$H$_4$ | Me | Me | **4e** | 87 | 96 |
| 8 | 3-MeSO$_2$C$_6$H$_4$ | Me | Me | **4f** | 85 | 96 |
| 9 | 2-ClC$_6$H$_4$ | Me | Me | **4g** | 67 | 96 |
| 10 | Ph | Me | Me | **4h** | 86 | 95 |
| 11[d] | 4-MeC$_6$H$_4$ | Me | Me | **4i** | 49 | 97 |
| 12 | 3-MeC$_6$H$_4$ | Me | Me | **4j** | 76 | 95 |
| 13 | 2-MeC$_6$H$_4$ | Me | Me | **4k** | 70 | 88 |
| 14[d] | 4-iBuC$_6$H$_4$ | Me | Me | **4l** | 41 | 97 |
| 15[d] | 3-MeOC$_6$H$_4$ | Me | Me | **4m** | 71 | 95 |
| 16 | 1-naphthyl | Me | Me | **4n** | 63 | 87 |
| 17 | 2-naphthyl | Me | Me | **4o** | 85 | 96 |
| 18[d] | 2-thienyl | Me | Me | **4p** | 49 | 96 |
| 19 | 2-Br pyrid-3-yl | Me | Me | **4q** | 65 | 89 |
| 20[d] | N-Ts 3-indolyl | Me | Me | **4r** | 45 | 94 |
| 21 | Cy | Me | Me | **4s** | 40 | 87 |
| 22 | 4-ClC$_6$H$_4$ | Pr | Me | **4t** | 69 | 91 |
| 23 | 4-ClC$_6$H$_4$ | iBu | Me | **4u** | 80 | 90 |
| 24 | 4-ClC$_6$H$_4$ | CH$_2$Ph | Me | **4v** | 61 | 95 |
| 25 | 4-ClC$_6$H$_4$ | Ph | Me | **4w** | 89 | 88 |
| 26 | 4-ClC$_6$H$_4$ | (CH$_2$)$_2$CO$_2$Me | Me | **4x** | 77 | 95 |
| 27 | 4-ClC$_6$H$_4$ | (CH$_2$)$_2$SMe | Me | **4y** | 45 | 89 |

[a]Reaction conditions: all reactions of step 1 were carried out with 0.3 mmol **1**, 0.2 mmol **2**, 0.3 mmol Cs$_2$CO$_3$ in 2 mL of CH$_2$Cl$_2$ for 10–14 h. Cu(I) = Cu(MeCN)$_4$BF$_4$. Ir(I) = [Ir(cod)Cl]$_2$; Step 2: 3.5 equiv TfOH in MeCN (0.1 M) at 40 °C for 2–3 days, >20:1 dr for all cases.
[b]Isolated yields over two steps.
[c]Ee was determined by chiral HPLC analysis.
[d]Performed at 50 °C.

**Fig. 3 | Gram-scale synthesis and derivatization of the cycloadduct 4a.** One-pot gram-scale synthesis and synthetic transformations.

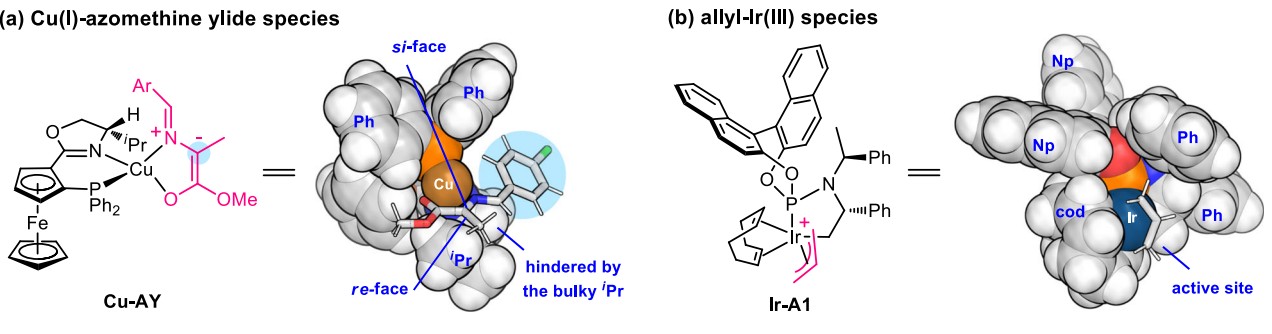

**(a) Cu(I)-azomethine ylide species**

**Cu-AY**

**(b) allyl-Ir(III) species**

**Ir-A1**

**(c) C–C coupling transition states between Cu(I)-ylide and allyl-Ir(III) with distortion/interaction and NCI analysis**

**TS1**
$\Delta G^{\ddagger} = 9.2$
producing (*S*)-**3a**
**Favored**

$\Delta E_{dis} = 17.0$
$\Delta E_{dis,Ir} = 15.3; \Delta E_{dis,Cu} = 1.7$
$\Delta E_{int} = -23.0$
$\Delta\Delta E^{\ddagger} = -6.0$

**VS**

$\Delta E_{dis} = 21.0$
$\Delta E_{dis,Ir} = 16.9; \Delta E_{dis,Cu} = 4.1$
$\Delta E_{int} = -21.0$
$\Delta\Delta E^{\ddagger} = 0.0$

**TS1'**
$\Delta G^{\ddagger} = 16.8$
producing (*R*)-**3a**
**Disfavored**

**Fig. 4 | Computational studies. a/b** The formation and structural analysis for Cu(I)-azomethine ylide **Cu-AY** and π-allyliridium(III) species **Ir-A1**. **c** The C−C coupling transition states for the allylation of Cu(I)-ylide **Cu-AY** by allyliridium(III) **Ir-A1** along with distortion/interaction analysis and non-covalent interaction analysis. Free energies are given in kcal/mol and relative to **Cu-AY** and **Ir-A1**, and selected bond distances are labelled in Å.

large aryl moiety points away from Cp-based and isopropyl substituents. In view of this large thermodynamic difference, Cu-ylide **Cu-AY'** will not be further pursued in the following procedures. Structural analysis for the diastereomeric prochiral nucleophile **Cu-AY** exhibits that its *si*-face was less hindered and active for electrophiles (Fig. 4a); while, its *re*-face was blocked by the oxazoline ring and its bulky isopropyl group. Accordingly, the subsequent allylation prefers to generate the (*S*)-compound (*see below*).

Meanwhile, the electrophilic π-allyl-Ir(III) complex was readily generated via decarboxylative oxidative addition (see Fig. 4b). DFT computational restults demonstrate that the coordination of allyl carbonate **1b** to Ir/**L5** and the ensuing decarboxylative step are facile to take place, and the active allyl-Ir(III) species **Ir-A1** was formed with thermodynamic driving force by 4.6 kcal/mol (see Supplementary Fig. 6). In the allyl-iridium(III) intermediate **Ir-A1**, structural examination reveals that the *back*-face was shielded by the cyclometallated moiety and cod ligand, and the *front*-face was unhindered to undergo

C-C coupling reaction. Afterward, the active catalysts gather together by way of a favored nucleophilic addition of the *si*-face of Cu-ylide **Cu-AY** to the *front*-face of π-allyl-Ir **Ir-A1** via **TS1** to finish alkylation and offer (*S*)-**3a** with an energy barrier of 9.2 kcal/mol (see Fig. 4c). By comparison, the nucleophilic attack between the *re*-face of Cu-ylide **Cu-AY** and the allyl-Ir species via **TS1′** (ΔG$^≠$ = 16.8 kcal/mol) has higher barrier than that of **TS1** (ΔG$^≠$ = 9.2 kcal/mol). This energy difference gives a calculated enantioselectivity (>99%) in favor of (*S*)-**3a** over (*R*)-**3a**, which agrees well with the experimental observation in Entry 10, Table 1. **TS1′** is disfavored mainly owing to that it involves strong steric repulsions between the cod ligand and the oxazoline ring and its bulky isopropyl substituent (see Fig. 4c). On the contrary, in **TS1**, it not only evades the strong ligand−ligand steric repulsions, but also is stabilized by the attractive C−H/π dispersion forces from the ligand−ligand and ligand−substrate interactions (see Fig. 4c). The differentiation in steric hindrance and dispersion effects could be clarified by distortion/interaction analysis[74], as labeled in Fig. 4c. Distortion/interaction energies demonstrate that it embraces less structural distortions ($E_{dis}$) in each catalyst (**Cu-AY** and **Ir-A1**) and greater interaction energies ($E_{int}$) between the two catalysts in **TS1** than that in **TS1′**. Coherently, non-covalent interactions (NCIs) analysis also unveils the differential forces[75]. Taken together, these analyses reasonably reveal the origins of stereoselectivity in the allylic alkylation stage.

After the generation of α-allylic substituted aldimine esters **3a**, it can not undergo rearrangement or cyclization reaction directly due to the high kinetic barrier (see Supplementary Fig. 7 for details). Herein, in the presence of Brønsted acid TfOH (see Fig. 5), the protonation of **3a** forms the adduct **Int-I**, in which the protonated imine can activate the electrophilic C=N bond and thus facilitate the cyclization via **TS2** (ΔG$^≠$ = 18.6 kal/mol). Note that, intrinsic reaction coordinate (IRC) calculations in both directions indicate that **TS2** connects complex **Int-I** and bridged intermediate **Int-II**. The IRC results suggest that the *aza*-Prins cyclization (C−C bond formation) and lactonization (C−O bond formation) occur simultaneously, which could be considered as a concerted process. Since the ester group serves as a nucleophile and completes the lactonization straightforwardly in the concerted procedure (**TS2**), other side reactions are effectively prohibited. The structural character of bridged intermediate **Int-II** hints that Me group has the cation feature and is able to be attacked by nucleophile to terminate the cyclization. This consideration lets us locate **TS3** leading to the final product **4a** and giving the by-product TfOMe. Moreover, compound (*S*)-**3a** may undertake chiral-inversion to form (*R*)-**3a**, which could endure similar reaction to give the stereoisomer of (1*S*,3*S*,5*R*)-**4a**. The computed isomerization pathway is exhibited in Fig. 5, which involves the ring-open (**TS4**), nucleophilic addition (**TS5**), N−C bond rotation, elimination (**TS6**) and *aza*-Cope rearrangement (**TS7**) to

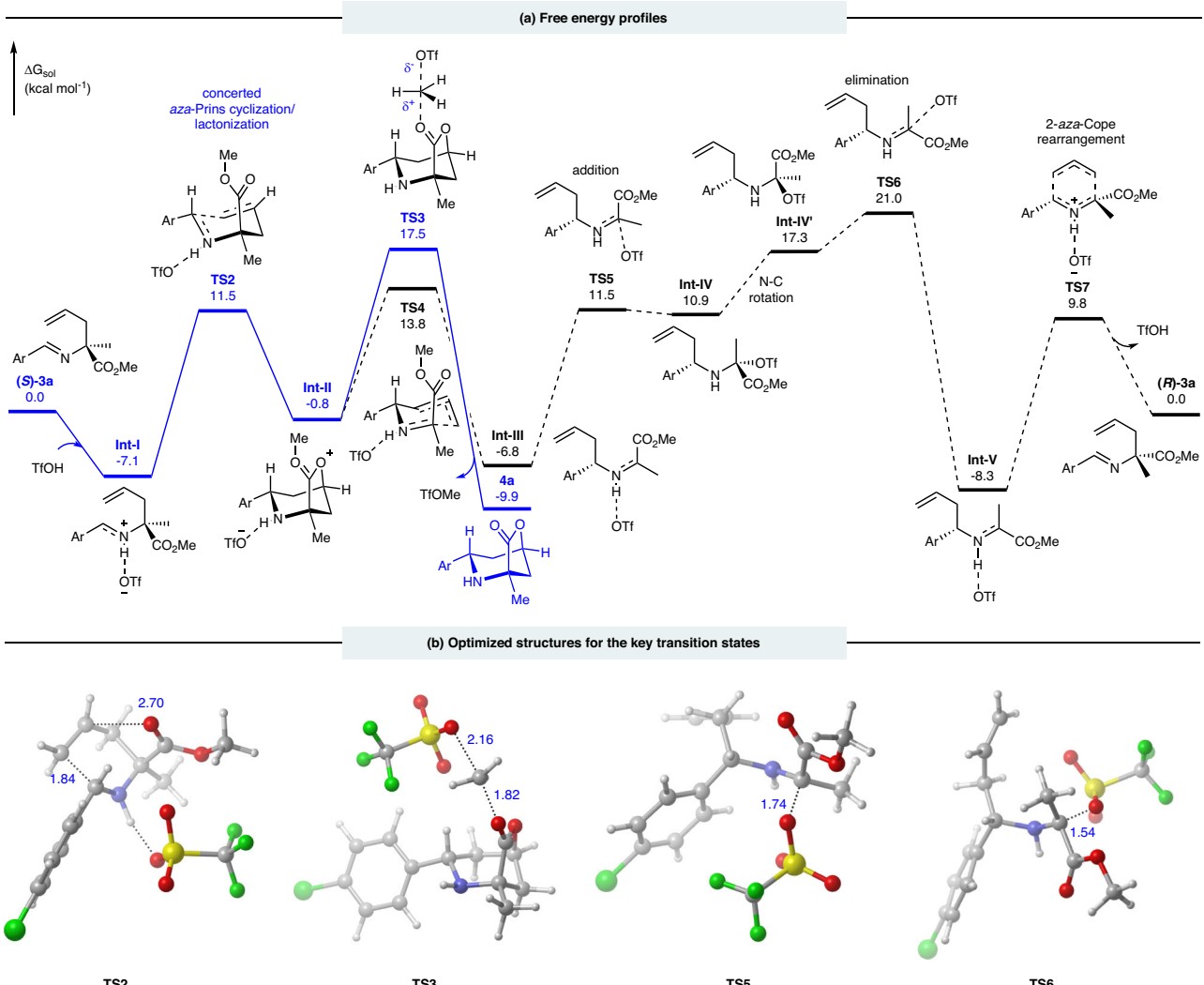

**Fig. 5 | Free energy profiles and optmized stuctures for the key transition states. a** Free energy profiles for the *aza*-Prins cyclization/lactonization and chiral-inversion processes. Free energies are given in kcal/mol; **b** Optmized structures for the key transition states with selected bond distances given in Å.

(a)

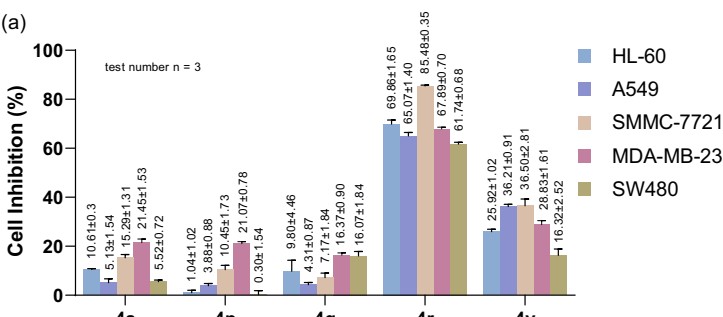

(b)

| cell lines | IC50 (μM) | |
|---|---|---|
| | **4r** | cisplatin |
| HL-60 | 0.114±0.032 | 14.15±0.75 |
| A549 | 0.680±0.081 | 18.00±0.39 |
| SMMC-7721 | 0.138±0.003 | 12.51±0.66 |
| MDA-MB-231 | 2.330±0.300 | 16.90±1.19 |
| SW480 | 0.280±0.018 | 25.06±1.26 |

**Fig. 6 | Biological activity study. a** 40 μM of five randomly selected products were incubated individually with HL-60 cancer cells, A549 lung carcinoma cells, SMMC-7721 hepatoma cells, MDA-MB-231 breast adenocarcinoma cells, and SW480 cells colon cancer cells (Data are presented as mean values ± SEM, test number $n = 3$).

**b** The cell viability after incubation with different concentrations of **4r** and their IC$_{50}$ values were determined in comparison to cisplatin (Data are presented as mean values ± SEM, test number = 5).

finish chiral-inversion. This isomerization process is 3.5 kcal/mol higher than the *aza*-Prins cyclization/lactonization pathway (**TS3** versus **TS6**), which makes the chiral-inversion kinetically unfavorable. The computed results are in agreement with the experimental observations and disclose the origins of stereoselectivity.

## Biological activity study

In Consideration of the important bioactive properties of piperidines and γ-butyrolactones, we are interested in the potential biological activities of these bridged piperidine-γ-butyrolactone heterocycles. Thus, the preliminary cytotoxic effects of five randomly selected bridged piperidine-γ-butyrolactones products were then investigated against a panel of cancer cell lines including HL-60 cancer cells, A549 lung carcinoma cells, SMMC-7721 hepatoma cells, MDA-MB-231 breast adenocarcinoma cells, and SW480 colon cancer cells by MTS assay (MTS, 3-(4,5-dimethylthiazol-2-yl)−5(3-carboxymethoxyphenyl)−2-(4-sulfopheny)−2H-tetrazolium). As shown in Fig. 6a, our preliminary studies revealed that compound **4r** displayed impressive cytotoxicity against these cancer cells at the concentration of 40 μM. In addition, the corresponding cytotoxicity of compound **4r** to these cancer cells with half maximal inhibitory concentration (IC$_{50}$) values in the low micromolar range were studied using the widely used anticancer drug cisplatin as the control, remarkably, it exhibited much lower IC$_{50}$ values (Fig. 6b). These preliminary results demonstrated that the distinctive bridged piperidine-γ-butyrolactone heterocycle owned the great potential for the development of new anti-cancer agents, and deserved further estimations for other bioactive properties.

## Discussion

In conclusion, we have developed a sequential process combined with bimetallic Cu/Ir-catalyzed asymmetric allylation, and *aza*-Prins cyclization/lactonization. This protocol provides an efficient entry to a variety of highly functionalized bridged piperidine-γ-butyrolactones in high yields with excellent diastereo-/enantios-electivity from readily-available starting materials. The synergistic Cu/Ir-catalyzed asymmetric allylation represents a unique example of asymmetric allylation using the simplest electrophilic π-allyl-Ir(III) species. The sequential construction of 2,4,6-three skipped stereogenic carbon centers with exclusive stereocontrol is another key feature of this process. Moreover, broad substrate scope, facile scale-up manipulation, and preliminary biological activities against a panel of cancer cell lines further demonstrated the great potential application value of this protocol. DFT mechanistic investigations construct a dual bimetallic catalytic cycle to rationalize the asymmetric allylation and further unveil an acid-aided concerted *aza*-Prins cyclization/lactonization to understand the consecutive annulations. Ligand effects with

steric repulsions and dispersion interactions were exposed to elucidate the origin of the stereoselectivity.

## Methods

### General reaction procedure

A flame-dried Schlenk tube was cooled to room temperature and filled with Ar. To this flask were added [Ir(COD)Cl]$_2$ (0.003 mmol, 1.5 mol%), phosphoramidite ligand (*S,S,S*)-**L5** (0.006 mmol, 3 mol%), degassed THF (0.5 mL) and degassed *n*-propylamine (0.5 mL). The reaction mixture was stirred at 50 °C for 30 min and then the volatile solvents were removed in vacuum to give a pale yellow solid. Meanwhile, in a separated Schlenk tube, (*S,S$_p$*)-*i*Pr-Phosferrox-**L1** (0.011 mmol, 5.5 mol%) and Cu(MeCN)$_4$BF$_4$ (0.01 mmol, 5 mol %) were dissolved in DCM (0.5 mL), and stirred at room temperature for about 0.5 h. A solution of aldimine ester **1** (0.30 mmol) in CH$_2$Cl$_2$ (0.5 mL) was added, followed by Cs$_2$CO$_3$ (98 mg, 0.30 mmol), allylic carbonate **4** (0.20 mmol) in DCM (0.5 mL) and pre-prepared Ir-complex in DCM (0.5 mL). The reaction mixture was stirred at 25 °C for 10–14 h. The organic solvent was removed by rotary evaporation. The residue can be used directly in ensuing acid-promoted *aza*-Prins cylization/lactonization or be purified by column chromatography (1–5% of EtOAc and 1% of Et$_3$N in PE) to afford the allylation product. To a solution of the obtained crude allylation product in MeCN (1.5 mL) was added TfOH (3.5 eq.), the reaction mixture was stirred at 40 °C for 2–3 days (Note: 50 °C was needed for some electron-enriched substrates, including the cases of **4i, 4 l, 4 m, 4p** and **4r**), basified with Et$_3$N (0.1 mL) and then concentrated in vacuum. The residue was purified by column chromatography on silica gel (PE:EtOAc = 6:1) afforded the bridged-heterocyclic product **4**. The dr value was determined by $^1$H NMR spectrum of the product, and the enantiomeric excess was recorded by HPLC analysis in comparison with the racemic sample.

### Reporting summary

Further information on research design is available in the Nature Portfolio Reporting Summary linked to this article.

## Data availability

Experimental procedures, characterization data, and source data are available within this article and its Supplementary Information. All other data are available from the corresponding author upon request. Cartesian coordinates of the calculated structures are available from the Supplementary Data 1. The X-ray crystallographic coordinates for the structures of compounds (1*S*,3*S*,5*R*)-**4a** and (1*R*,3*R*,5*S*)-**4a** reported in this study have been deposited at the Cambridge Crystallographic Data Centre (CCDC), under deposition numbers 2256426 and 2256427. These data can be obtained free of charge from The Cambridge Crystallographic Data Centre via www.ccdc.cam.ac.uk/data_request/cif.

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

## Acknowledgements

This work was supported by National Key R&D Program of China (2023YFA1506700 (C.J.W.)), NSFC (22071186 (C.J.W.), 22071187 (X.Q.D.), 22073067 (Y.D.), 22101216 (X.C.), 22271226 (X.Q.D.), and 22371216 (C.J.W.)), National Youth Talent Support Program, Hubei Province NSF (2020CFA036 (C.J.W.) and 2021CFA069 (X.Q.D.)), and China Postdoctoral Science Foundation (2021M702514 (X.C.)), and Fundamental Research Funds for the Central Universities (2042022kf1180 (C.J.W.), and 2042022kf1040 (Z.Z.)). The authors thank Dr. Ran Zhang from the Core Facility of Wuhan University for his generous support in the X-ray structures analysis, and Hongmei Li and Xingzhi Yang from Natural Drug Activity Screening Center in Kunming Institute of Botany of Chinese Academy of Sciences for their generous support in the cytotoxicity study.

## Author contributions

C.J.W. conceived and designed the research. C.J.W. and X.Q.D. directed the project. C.F., L.H., X.C and Z.Z performed the research. Y.D. directed the DFT calculation, and H.X. performed the DFT calculation research. C.F., Y.D., X.Q.D. and C.J.W. co-wrote the paper. All authors analyzed the data, discussed the results, and commented on the manuscript.

## Competing interests

The authors declare no competing interests.
