## [Peer Review File · Nature Communications]

REVIEWER COMMENTS

Reviewer #1 (Remarks to the Author):

It is well-known that chiral functionalized piperidine and lactone heterocycles are valuable scaffolds in many natural products and drug candidates with promising pharmacological properties. Until now, there is no general and efficient synthetic methods to prepare these important chiral molecules. In this research work, Wang and coworkers developed a facile relay strategy for the construction of bridged chiral piperidine- γ -butyrolactone skeletons incorporating three skipped stereocenters through Cu/Ir-catalyzed asymmetric allylic alkylation and acid-promoted aza-Prins cyclization/lactonization sequences. A series of heterocyclic bridged piperidine- γ -butyrolactones with three skipped stereocenters could be readily obtained in high yields with excellent diastereo-/enantioselective control. Moreover, preliminary biological activities investigation showed that they are effective against a panel of cancer cell lines with impressive and promising results. In addition, systematic DFT studies were carried out to reveal a dual bimetallic catalytic cycle to rationalize the asymmetric allylation and further unveil an acid-aided concerted aza-Prins cyclization/lactonization. This research work is very interesting and important. The manuscript is generally well written and SI is well presented. Therefore, I recommend publication of this paper in Nature Communications, and only have the following suggestions.

- (1) Table 1, "1b (PG = OCO₂Me)" should be as "1b (PG = CO₂Me)"
- (2) For aldimine esters, only methyl ester was used in the optimization, the authors should examine the corresponding ethyl ester and tert-butyl ester. Especially, tert-butyl ester maybe much easier for lactonization step due to the formation of isobutene.
- (3) What happens with the aldimine substrate derived from natural valine (R' = iPr) and unnatural tert-leucine (R' = tBu)? If those substrates are not tolerated in the transformation due to steric hindrance, the authors should mention the limitations in the context.
- (4) CDCl₃ is NMR solvent for the analysis of new compounds, and it is a common practice to reference the chemical shifts of the residual proton resonance of CHCl₃ (δ = 7.26 ppm) (Organometallics 2010, 29, 2176-2179). The authors should check all the ¹H NMR spectra, for example, compound 4f, the 7.29 ppm should be 7.26 ppm.

Reviewer #2 (Remarks to the Author):

The authors tested the anti-proliferative activity for five randomly selected bridged piperidine- γ -butyrolactones products against 5 cancer cell lines including HL-60 cancer cells, A549 lung carcinoma cells, SMMC-7721 hepatoma cells, MDA-MB-231 breast adenocarcinoma cells, and SW480 colon cancer cells by MTS assay. Compound 4r exhibited the highest anti-proliferative activity with IC50 of 0.114~2.330 μ M, significantly better than the positive control cisplatin (14.15~25.06 μ M). How about the cytotoxicity against normal cells?

Reviewer #3 (Remarks to the Author):

In this manuscript, Wang and coworkers described a new strategy involving synergistic Cu/Ir-catalyzed asymmetric allylation/aza-Prins cyclization/lactonization for the construction of enantioenriched bridged piperidine- γ -butyrolactone frameworks. The protocol features excellent diastereo-/enantioselectivity control, broad substrate scope and mild conditions. Compared with the previous works (ref. 39 and 69) by the same group, it's quite interesting that different types of reactions can occur with different substituted substrates under the same synergistic Cu/Ir catalysis. In addition, the obtained chiral piperidine- γ -butyrolactone bridged heterocyclic products showed good cytotoxicity against several cancer cells in the preliminary biological activity studies. This reviewer supports acceptance of this manuscript for publication in Nature Communications after some revisions.

1. Comparing the results of entry 10 and 11 in Table 1, as well as the DFT mechanism studies, it can be found that the enantioselectivity of the product is dominantly controlled by the chiral environment of the copper complex. For the metal iridium catalyst, perhaps the chiral ligand is not necessary, have the authors tried using racemic L5 or other simpler and cheaper chiral ligands?
2. The aza-Prins cyclization/lactonization process is quite slow (takes 2-3 days), from the free energy profile shown in Figure 5A, it can be seen that the methyl leaving step is in high energy state. One might wonder if the lactonization would be easier by changing methyl ester to tert-butyl ester or COOH.
3. Have the substituted allyl carbonate substrates ever been tested in this reaction sequence? What happens if the products obtained from substituted allylic methyl ester as reported in ref.39 (JACS 2018, 140, 1508) are treated with the same acid conditions instead of NaBH₃CN reduction?
4. In Table 1, 1b (PG = OCO₂Me) should be corrected to 1b (PG = CO₂Me).
5. For references 42, 44, 46, 61, 62, all authors should be listed to maintain consistency with other references.

Reviewer #4 (Remarks to the Author):

In this manuscript, Wang, Dong, and Dang et al. successfully realized dual Cu/Ir-catalyzed asymmetric allylic alkylation and acid-promoted aza-Prins cyclization/lactonization sequences, and a wide range of chiral bridged piperidine- γ -butyrolactone heterocycles were obtained in good yields with excellent diastereoselectivities and enantioselectivities (up to 96% yield, >20:1 dr, 97% ee). It may be that the features of three-skipped stereogenic centers with excellent stereoselectivity controls in aza-Prins cyclization/lactonization are even more impressive. Gram-scale synthesis and various synthetic transformations displayed potential application of this protocol. Moreover, DFT studies were conducted to explain the possible reaction mechanism and the stereochemical outcomes of this sequential transformation. It does appear that these authors have achieved a measure of success in aza-Prins cyclization reaction that has not yet been well-developed. Additionally, biological activities of the representative heterocyclic molecules were investigated with some preliminarily positive results. Thus, this work is an important study, and the manuscript was well-written. Overall, the significance of chiral bridged piperidine- γ -butyrolactone skeletons and the novelty and elegance of the straightforward relay strategy via bimetallic catalyst and acid render the current manuscript of significant interest to the readership of Nat. Commun.. I recommend publishing this work in Nat. Commun. after addressing the following minor additions/corrections.

(1) The authors tested allyl carbonates with OBoc and OCO₂Me as the leaving groups, how about the results using allyl acetate (OAc as the leaving group) as the precursor of Ir- π -allyl intermediate?

(2) For Ir-catalyzed asymmetric allylic alkylation, Feringa's type of chiral phosphoramidite L5 was employed. The authors should examine Carreira's type of chiral phosphoramidite (P, olefin) ligand, which is also commonly-used in Ir-catalyzed asymmetric allylic alkylation.

(3) In the DFT-computed pathway, the anionic TfO⁻ was suggested to aid the leaving of Me⁺. Has the author considered the influence of solvent (i.e., MeCN)?

Reviewer #1

Q1: *It is well-known that chiral functionalized piperidine and lactone heterocycles are valuable scaffolds in many natural products and drug candidates with promising pharmacological properties. Until now, there is no general and efficient synthetic methods to prepare these important chiral molecules. In this research work, Wang and coworkers developed a facile relay strategy for the construction of bridged chiral piperidine- γ -butyrolactone skeletons incorporating three skipped stereocenters through Cu/Ir-catalyzed asymmetric allylic alkylation and acid-promoted aza-Prins cyclization/lactonization sequences. A series of heterocyclic bridged piperidine- γ -butyrolactones with three skipped stereocenters could be readily obtained in high yields with excellent diastereo-/enantioselective control. Moreover, preliminary biological activities investigation showed that they are effective against a panel of cancer cell lines with impressive and promising results. In addition, systematic DFT studies were carried out to reveal a dual bimetallic catalytic cycle to rationalize the asymmetric allylation and further unveil an acid-aided concerted aza-Prins cyclization/lactonization. This research work is very interesting and important. The manuscript is generally well written and SI is well presented. Therefore, I recommend publication of this paper in Nature Communications, and only have the following suggestions.*

A1: Thank you very much for your great appraisal. We revised our manuscript and supporting information according to your following suggestions.

Q2: *Table 1, “1b (PG = OCO₂Me)” should be as “1b (PG = CO₂Me)”*

A2: Thank you very much for your kind reminder, it was revised in the manuscript.

Q3: *For aldimine esters, only methyl ester was used in the optimization, the authors should examine the corresponding ethyl ester and tert-butyl ester. Especially, tert-butyl ester maybe much easier for lactonization step due to the formation of isobutene.*

A3: Thank you for your kind suggestion. The corresponding ethyl and tert-butyl aldimine ester have been examined in this reaction. When the methyl group of aldimine ester was switched to ethyl ester, product **4a** was obtained in moderate yield with unsatisfactory enantioselectivity (Table 2, entry 2). Bulky tert-butyl aldimine ester was further examined, and the desired product

could be accessible in almost similar results (entry 3), and methyl aldimine ester was selected for subsequent investigation due to the high cost of *tert*-butyl aldimine esters. These experimental results have been added in Table 2 as entries 2 and 3 in the revised manuscript.

Q4: *What happens with the aldimine substrate derived from natural valine (R' = *i*Pr) and unnatural *tert*-leucine (R' = *t*Bu)? If those substrates are not tolerated in the transformation due to steric hindrance, the authors should mention the limitations in the context.*

A4: Thank you very much for your kind suggestion. Due to the steric hindrance, no reaction occurred with the aldimine esters derived from valine (R' = *i*Pr) and unnatural *tert*-leucine (R' = *t*Bu) under the standard reaction conditions. These experimental results have been described in the revised manuscript.

Q5: *CDCl₃ is NMR solvent for the analysis of new compounds, and it is a common practice to reference the chemical shifts of the residual proton resonance of CHCl₃ ($\delta = 7.26$ ppm) (Organometallics 2010, 29, 2176-2179). The authors should check all the ¹H NMR spectra, for example, compound 4f, the 7.29 ppm should be 7.26 ppm.*

A5: Thank you very much for your kind suggestion. The mention mistakes have been corrected in the revised Supplementary Information.

Reviewer #2

Q: *The authors tested the anti-proliferative activity for five randomly selected bridged piperidine- γ -butyrolactones products against 5 cancer cell lines including HL-60 cancer cells, A549 lung carcinoma cells, SMMC-7721 hepatoma cells, MDA-MB-231 breast adenocarcinoma cells, and SW480 colon cancer cells by MTS assay. Compound 4r exhibited the highest anti-proliferative activity with IC₅₀ of 0.114~2.330 M, significantly better than the positive control cisplatin (14.15~25.06 M). How about the cytotoxicity against normal cells?*

A: Thank you very much for your kind suggestion. The cytotoxicity of **4r** against normal cells L02 and BEAS-2B was examined, and the results were summarized as following:

compound	L02	BEAS-2B
	IC ₅₀ ± SD (μM)	
4r	0.167 ± 0.012	0.542 ± 0.026
cisplatin	12.00 ± 0.34	36.15 ± 0.99
Taxol	<0.008	0.765 ± 0.028

Reviewer #3

Q1: *In this manuscript, Wang and coworkers described a new strategy involving synergistic Cu/Ir-catalyzed asymmetric allylation/aza-Prins cyclization/lactonization for the construction of enantioenriched bridged piperidine-γ-butyrolactone frameworks. The protocol features excellent diastereo-/enantioselectivity control, broad substrate scope and mild conditions. Compared with the previous works (ref. 39 and 69) by the same group, it's quite interesting that different types of reactions can occur with different substituted substrates under the same synergistic Cu/Ir catalysis. In addition, the obtained chiral piperidine-γ-butyrolactone bridged heterocyclic products showed good cytotoxicity against several cancer cells in the preliminary biological activity studies. This reviewer supports acceptance of this manuscript for publication in Nature Communications after some revisions.*

A1: Thank you very much for your great appraisal. We revised our manuscript and supporting information according to your following suggestions.

Q2: *Comparing the results of entry 10 and 11 in Table 1, as well as the DFT mechanism studies, it can be found that the enantioselectivity of the product is dominantly controlled by the chiral environment of the copper complex. For the metal iridium catalyst, perhaps the chiral ligand is not necessary, have the authors tried using racemic L5 or other simpler and cheaper chiral ligands?*

A2: Thank you very much for your kind suggestion. Unsatisfactory asymmetric induction was observed when the combined [Cu(I)/(S,S_p)-L1 + Ir(I)/rac-L5] was employed in this allylation (Table 1, entry 13). In addition, the (S)-Carreira's ligand was also applied, and product **3a** was obtained in good yield albeit with unsatisfactory enantioselectivity (Table 1, entries 14 and 15).

These experimental results have been added in Table 1 as entries 13-15 in the revised manuscript.

Q3: *The aza-Prins cyclization/lactonization process is quite slow (takes 2-3 days), from the free energy profile shown in Figure 5A, it can be seen that the methyl leaving step is in high energy state. One might wonder if the lactonization would be easier by changing methyl ester to tert-butyl ester or COOH.*

A3: Thank you for your kind suggestion. When the methyl group of aldimine ester was switched to *tert*-butyl ester, a little higher reactivity was observed and the desired product **4a** could be accessible in almost similar yield and enantioselectivity (Table 2, entry 3). Methyl aldimine ester was selected for subsequent investigation due to the high cost of *tert*-butyl aldimine esters. These experimental results have been added in Table 2 as entry 3 in the revised manuscript.

Q4: *Have the substituted allyl carbonate substrates ever been tested in this reaction sequence? What happens if the products obtained from substituted allylic methyl ester as reported in ref.39 (JACS 2018, 140, 1508) are treated with the same acid conditions instead of NaBH₃CN reduction?*

A4: Thank you very much for your kind suggestion. The allylation intermediates from the substituted allyl carbonates was easily decomposed in the same acid conditions, and no *aza*-Prins cyclization/lactonization reaction occurred.

Q5: *In Table 1, 1b (PG = OCO₂Me) should be corrected to 1b (PG = CO₂Me).*

A5: Thank you very much for your kind suggestion. It was corrected in the revised manuscript.

Q6: *For references 42, 44, 46, 61, 62, all authors should be listed to maintain consistency with other references.*

A6: Thank you very much for your kind suggestion. We listed all authors for references 42, 44, 46, 61 and 62 in the revised manuscript.

Reviewer #4

Q1: *In this manuscript, Wang, Dong, and Dang et al. successfully realized dual Cu/Ir-catalyzed asymmetric allylic alkylation and acid-promoted aza-Prins cyclization/lactonization sequences,*

and a wide range of chiral bridged piperidine- γ -butyrolactone heterocycles were obtained in good yields with excellent diastereoselectivities and enantioselectivities (up to 96% yield, >20:1 dr, 97% ee). It may be that the features of three-skipped stereogenic centers with excellent stereoselectivity controls in aza-Prins cyclization/lactonization are even more impressive. Gram-scale synthesis and various synthetic transformations displayed potential application of this protocol. Moreover, DFT studies were conducted to explain the possible reaction mechanism and the stereochemical outcomes of this sequential transformation. It does appear that these authors have achieved a measure of success in aza-Prins cyclization reaction that has not yet been well-developed. Additionally, biological activities of the representative heterocyclic molecules were investigated with some preliminarily positive results. Thus, this work is an important study, and the manuscript was well-written. Overall, the significance of chiral bridged piperidine- γ -butyrolactone skeletons and the novelty and elegance of the straightforward relay strategy via bimetallic catalyst and acid render the current manuscript of significant interest to the readership of Nat. Commun.. I recommend publishing this work in Nat. Commun. after addressing the following minor additions/corrections.

A1: Thank you very much for your great appraisal. We revised our manuscript and supporting information according to your following suggestions.

Q2: *The authors tested allyl carbonates with OBoc and OCO₂Me as the leaving groups, how about the results using allyl acetate (OAc as the leaving group) as the precursor of Ir- π -allyl intermediate?*

A2: Thank you for your kind suggestion. Allyl acetate has been examined in this reaction, and the desired product **3a** was obtained in 84% yield with 93% ee. These experimental results have been added in Table 1 as entry 11 in the revised manuscript.

Q3: *For Ir-catalyzed asymmetric allylic alkylation, Feringa's type of chiral phosphoramidite L5 was employed. The authors should examine Carreira's type of chiral phosphoramidite (P, olefin) ligand, which is also commonly-used in Ir-catalyzed asymmetric allylic alkylation.*

A3: Thank you very much for your kind suggestion. The Carreira's ligand has been applied, and product **3a** was obtained in good yield albeit with unsatisfactory enantioselectivity (Table 1,

entries 14 and 15). These experimental results have been added in Table 1 as entries 13-15 in the revised manuscript

Q4: In the DFT-computed pathway, the anionic TfO⁻ was suggested to aid the leaving of Me⁺. Has the author considered the influence of solvent (i.e., MeCN)?

A4: We have considered the leaving of Me⁺ moiety assisted by MeCN solvent. DFT computational result demonstrates that this type reaction (**TS3-NCMe**) is 11.9 kcal/mol higher than that of **TS3**, excluding the reaction possibility. We have added the computed result in Figure 3B.

Figure S3. (A) Free energy profiles for the direct *aza*-Cope rearrangement of (*S*)-**3a**. (B) Transition states for the leaving of Me⁺ group aided by MeCN. The TfO⁻ moiety was omitted for clarity in the optimized structure of **TS3-CNMe**.

REVIEWERS' COMMENTS

Reviewer #1 (Remarks to the Author):

All the comments have been responded or corrected , and this reviewer recommend publication as it is.

Reviewer #2 (Remarks to the Author):

The authors have addressed my comments and the paper can be accepted.

Reviewer #3 (Remarks to the Author):

The authors have addressed most of my points, I would suggest this work to be accepted for publicationbe in Nature Communications.

Reviewer #4 (Remarks to the Author):

Authors have answered my concerns, so it could be publishable.